# A randomised controlled trial of email versus mailed invitation letter in a national longitudinal survey of physicians

**Benjamin Harrap[1], Tamara Taylor[2], Grant Russell[3], Anthony Scott[4]***

**1** Centre for Health Policy, Melbourne School of Population and Global Health, The University of Melbourne, Carlton, Victoria, Australia, **2** Government and Social Research Division, Big Village, Melbourne, Australia, **3** Department of General Practice, Monash University, Melbourne, Victoria, Australia, **4** Centre for Health Economics, Monash University, Caulfield East, Victoria, Australia

* anthony.scott@monash.edu

**Data Availability Statement:** Data can requested using the following non-author email: mabel-admin@unimelb.edu.au at the Melbourne Institute: Applied Economic and Social Research at the

## Abstract

Despite their low cost, the use of email invitations to distribute surveys to medical practitioners have been associated with lower response rates. This research compares the difference in response rates from using email approach plus online completion rather than a mailed invitation letter plus a choice of online or paper completion. A parallel randomised controlled trial was conducted during the 11th annual wave of the nationally representative Medicine in Australia: Balancing Employment and Life (MABEL) longitudinal survey of doctors. The control group was invited using a mailed paper letter (including a paper survey plus instructions to complete online) and three mailed paper reminders. The intervention group was approached in the same way apart from the second reminder when they were approached by email only. The primary outcome is the response rate and the statistical analysis was blinded. 18,247 doctors were randomly allocated to the control (9,125) or intervention group (9,127), with 9,108 and 9,107 included in the analysis. Using intention to treat analysis, the response rate in the intervention group was 35.92% compared to 37.59% in the control group, a difference of -1.66 percentage points (95% CI: -3.06 to -0.26). The difference was larger for General Practitioners (-2.76 percentage points, 95% CI: -4.65 to -0.87) compared to other specialists (-0.47 percentage points, 95% CI: -2.53 to 1.60). For those who supplied an email address, the average treatment effect on the treated was higher at -2.63 percentage points (95% CI: -4.50 to -0.75) for all physicians, -3.17 percentage points (95% CI: -5.83 to -0.53) for General Practitioners, and -2.1 percentage points (95% CI: -4.75 to 0.56) for other specialists. For qualified physicians, using email to invite participants to complete a survey leads to lower response rates compared to a mailed letter. Lower response rates need to be traded off with the lower costs of using email rather than mailed letters.

## Background

Web surveys have consistently lower response rates than all other survey modes [1]. Surveys of medical practitioners remain a key source of information about clinical practice, health service delivery, and clinical attitudes and experience. A key issue with survey data is that they can

University of Melbourne. In addition, data can be requested from the Australian Data Archive https://dataverse.ada.edu.au/dataverse/mabel. This provides access to main MABEL survey data but not the data used in this trial.

**Funding:** This research used data from the MABEL longitudinal survey of doctors. Funding for MABEL was provided by the National Health and Medical Research Council (2007 to 2016: 454799 and 1019605); the Australian Department of Health and Ageing (2008); Health Workforce Australia (2013); The University of Melbourne, Medibank Better Health Foundation, the NSW Department of Health, and the Victorian Department of Health and Human Services (2017); and the Australian Government Department of Health, the Australian Digital Health Agency, and the Victorian Department of Health and Human Services (2018). The funders had no role in study design, data collection and analysis, decision to publish, or preparation of the manuscript. AS was the Chief Investigator for all above grants but did not receive salary or payments. BH and TT were employed using the above research funding. GR did not receive salary or payments.

**Competing interests:** The authors have declared that no competing interests exist.

have low external validity because the sample may be less representative due to response bias caused by recruitment methods and non-random selection of physicians completing the survey. Although a low response rate does not necessarily mean low external validity [2], the focus on response rates remains a key feature of the survey methods literature for physicians [3–5].

Systematic reviews and meta-analyses have examined different methods of increasing response rates in surveys of medical practitioner populations [3,6–8], such as changing features of survey design and delivering incentives. Email contact and online survey completion is popular as costs are lower but research has shown that response rates also tend to be lower, with a mailed approach more effective and recommended [3]. For example, in a meta-analysis of 48 studies of health professionals, three studies found that mailed surveys were associated with higher response rates than online/web modes, with no difference in response rates between online modes and mixed modes [3]. Pit, Vo (6) conducted a systematic review of methods used to increase response rates for GPs, and found postal surveys were more effective than phone or email surveys (as a singular method of distribution), and a sequential mixed-mode of reminders was more effective than using online only or online and paper surveys concurrently. Beebe, Jacobson [9] found that a sequential mixed mode (web followed by mail) survey of health professionals had a higher response rate than mail only but found no statistically significant differences between mail only and web only, though the sample sizes were small. No differences were found between web only, mail only, and mixed modes in a more recent randomised study of physicians [10]. Other key studies have examined mixed modes that compare combinations of mail and online approaches, but do not directly compare mail and online [11–13].

Most studies used data generally more than 10 years old. As the use of email and the internet becomes more universal, including the more widespread use of electronic medical records, it is important to re-examine this issue. Nevertheless, for physician cohorts who are less familiar with the internet, mainly older physicians, there is uncertainty as to whether response rates would be different, and a risk that response rates will be lower with an email approach or online completion. Older people are less likely to respond to email than younger people [14,15] and if a mailed approach is used and physicians are given the choice between paper or online competition, the latter is less likely for older physicians [16].

The aim of this research is to compare response rates between an email approach and a mailed approach within a national longitudinal survey of physicians. More specifically, we introduce an email approach in the second of three reminders sent to non-responding physicians. In the first ten annual waves of the survey, the main mailout and all three reminders were delivered by mail only. Our null hypothesis was that there would be no difference in the response rate when delivering the reminder by email or mail.

## Methods

Reporting and design of the randomised trial are based on the Consolidated Standards of Reporting Trials (CONSORT) guidelines [17]. The study was approved by The University of Melbourne Faculty of Business and Economics Human Ethics Advisory Group (Ref. 0709559) and the Monash University Standing Committee on Ethics in Research Involving Humans (Ref: 195535 CF07/1102–2007000291). Participant consent was obtained through voluntarily completing the survey.

### Participants

The research was conducted within the context of the Medicine in Australia: Balancing Employment and Life (MABEL) survey. This was a longitudinal panel survey of all medical

practitioners in Australia, collecting 11 annual waves of data from around 9,000 to 10,000 physicians per wave [18]. The original responders in Wave 1 (2008) were followed up annually, with the addition of a cohort of new doctors entering the sample frame from Wave 2 and each subsequent wave [19]. Each wave therefore had a mixture of doctors from different cohorts. Responses for each wave were gathered using a sequential mixed mode design based on an earlier RCT [11]. The MABEL survey is sent to all types of medical practitioners in Australia.

The sample frame for MABEL is the Medical Directory of Australia, a national database of doctors held by the Australasian Medical Publishing Company (AMPCo). We use participants from Wave 11, administered between August 2018 and April 2019. Doctors were excluded if they had previously requested to withdraw from the MABEL survey, or who were known to be deceased. Junior doctors were excluded since in 2016 we conducted a small experiment that supported the use of an email approach for junior doctors and this was adopted in subsequent waves for this group only [18].

The invitation included a mailed letter that contained unique log in details for online completion to enable longitudinal tracking, as well as a paper copy of the survey which included a unique username printed on the cover. Respondents could choose the mode of completion. The first reminder used a mailed paper letter containing instructions for online completion but no paper survey. The second reminder used a mailed letter with instructions for online completion and included a paper copy of the survey. The third reminder included only a mailed paper letter with instructions for online completion.

## Intervention

The mailout for the intervention groups included an email approach for the second reminder. Both the intervention and control group were approached four times: the initial invitation plus three reminders. In the control group all four approaches used a mailed paper letter sent to each participant's work address. All survey materials are available at www.mabel.org.au.

The intervention group only differed at the second reminder, where they were approached by email and could only complete online, receiving no paper letter or paper copy of the survey. The comparison between the intervention and control group therefore includes a different method of approach and a different method of completion: email approach plus online completion versus mailed approach plus a choice of online or paper completion. The email included the same text as the paper letter plus a link and instructions for online completion. Emails were sent to email addresses from the AMPCo database or from email addresses provided by participants in earlier waves of the survey.

## Outcomes

The primary outcome for this study is the response rate at end of recruitment. A medical practitioner was considered to have responded if they returned their survey (via mail or completing online) with Section A completed which included questions on whether they were currently participating in clinical practice and at least one question answered from Section B. Surveys returned blank were counted as refusals to participate. The response rate was calculated as the total number of responses divided by the total number of surveys distributed (minus surveys that could not be sent by the mailing house because the doctor was deceased or had no valid mailing address).

## Sample size

The sample for the trial included 18,247 GPs and non-GP specialists eligible to be invited to complete a survey in Wave 11. This included, i) 13,382 doctors who had previously completed

at least one MABEL survey since 2008 (defined as a continuing doctor), ii) a cohort of 1,862 doctors new to the sample frame in 2018, and iii) 3,003 doctors from a 'boost' sample comprising a 10% random sample of those who had never responded to an invitation to participate in MABEL. The total sample size of 18,247 doctors is sufficient to detect a two-sided difference of least two percentage points in the response rate (alpha 0.05 and power 0.8). This assumes a response rate of 42.4% in the control group (an estimate from Wave 10 of MABEL).

## Randomisation

A parallel-arm design with 1:1 allocation was used, with 18,247 doctors randomly allocated to either the control or intervention group. Allocation was stratified by doctor type (GP, specialist), continuing or new, and boost sample to ensure the proportions of these groups of doctors in the intervention and control groups were the same. This is important because new doctors and boost sample doctors are likely to have lower response rates, and specialists had higher response rates than GPs in previous waves. We tested a two-sided hypothesis as it was unclear, a priori, whether the intervention group would have a higher or lower response rate. Randomisation was performed using the *sample* command in Stata 15.1 statistical software[15].

Randomisation took place (by TT) before the first invitation for Wave 11 was mailed out in August 2018. Group allocation was kept separate (in a separate electronic data file) from the main mailout and the first reminder so researchers handling the responses and reminders were blinded to group allocation during this process. The second reminder was prepared in late November 2018 by TT. The list of those eligible for a second reminder was then merged with the file containing the intervention and control group identifiers to indicate who should receive an email. A separate file indicating whether doctors had an email address was also merged onto this file. AMPCo was sent a list of doctor identifiers indicating if they should be approached using a mailed letter or an email.

## Statistical methods

The analysis was conducted by BH who was blinded to group allocation until after the analysis was complete and checked, and who was not involved in the randomisation or any data collection. Baseline characteristics of the intervention and control groups are compared with each other, and with the population of medical practitioners in Australia. The main analysis was based on intention to treat, which estimates the average treatment effect (ATE).

The proportions responding in each group were compared using a 2x2 table and Pearson chi-squared test. An adjusted analysis was also conducted using multivariable logistic regression to examine the probability of response between the two groups after adjusting for covariates. Sub-group analysis was also conducted using separate logistic regressions for GPs and other specialists. Covariates included age, gender, whether qualified overseas, quartiles of the socio-economic status of patients in each respondent's postcode (measured using the Socio-Economic Indexes For Areas (SEIFA) Index of Relative Socio-Economic Disadvantage [20]), and the proportion of the population in the postcode over 65 years old and under 5 years old. Finally, the rurality of work location was measured using the Modified Monash Model (MMM) classification [21]. Major cities (MMM1), areas within 20km of town with 50,000 population (MMM2); areas within 15km of town with 15,000 to 50,000 population (MMM3); areas within 10km of town with 5,000 to 15,000 population (MMM4); MMM5-7 (all other remote and rural areas) are grouped with MM4 for the analysis. Statistical analysis was conducted using STATA [22].

A proportion of doctors allocated to the intervention and control group did not supply a valid email address to AMPCo. These will be allocated equally across intervention and control

groups. Those who did not supply an email address in the intervention group cannot adhere to their allocated group and were instead approached by mailed paper letter rather than email. The intention to treat analysis includes these in the intervention group. This is appropriate as it assumes that in practice not all doctors are willing to provide email addresses and so the main results are applicable to the population of doctors whether or not they are willing to supply an email address.

However, this will lead to an underestimate of the effect of the intervention for those who actually received an email compared to those in the control group who also received email. In addition to calculating the results using intention to treat analysis, we therefore calculate the average treatment effect on the treated (ATET). This compares those who had a valid email address in both the intervention and control group.

## Results

A comparison of the sample used in the trial and the population of GPs and specialists in clinical practice in 2018 shows that the trial sample was more likely to be female, are slightly younger, less likely to be from New South Wales, more likely to be from a non-metropolitan area. The proportion who are a specialist, the socio-economic status of the population, and the proportion of the population aged under 5 and over 65 years old are similar (Table 1). Descriptive statistics comparing the characteristics of the intervention and control groups are shown in Tables 2 and 3. The flow diagram in Fig 1 shows each step of the study and how the final sample was determined. Comparison of response rates are shown in Table 4, overall and for the subgroups of GPs and non-GP specialists. The response rate in the intervention group was 35.92% compared to 37.59% in the control group, a difference of -1.66 percentage points (95% CI: -3.06 to -0.26). After adjustment for covariates, this increases to -1.93 (95% CI: -3.36 to -0.50). The difference was larger for GPs (-2.76 percentage points, 95% CI: -4.65 to -0.87) compared to non-GP specialists (-0.47 percentage points, 95% CI: -2.53 to 1.60).

The estimates of the ATET are shown in the bottom half of Table 3. This analysis compares only those who were approached by email in the intervention group to those in the control who had supplied an email address but were approached by mailed letter. Of those who were sent a second reminder in the intervention group, 43% (2840/6605) did not have an email address compared to 43.8% (2866/6540) in the control group. The overall ATET is larger compared to the ITT effect (-2.63 percentage points, 95% CI: -4.50 to -0.75), as well as for GPs (-3.17 percentage points, 95% CI: -5.83 to -0.53) and specialists (-2.10 percentage points, 95% CI: -4.75 to 0.56). The overall difference falls to 2.37 percentage points after adjustment for covariates.

## Discussion

Using email to approach potential survey subjects is often preferred because of its low cost. It is likely to have gained popularity during the COVID-19 pandemic given issues with collecting survey data using face-to-face interviews. Several studies have shown that in surveys of physicians an emailed approach can lead to lower response rates, potentially increasing response bias and reducing external validity. This is also the case in other populations [1]. However, these studies are of specific samples and most are over 10 years old. One might assume that since then the use of email, familiarity with the internet, and online survey completion should have become more familiar to physician populations.

Our results confirm that, in a nationally representative population of qualified GPs and non-GP specialists, response rates are lower using an emailed approach. The finding of a 1.93 percentage point fall in the response rate is from the ITT analysis and so is relevant to

**Table 1. Comparison of trial participants with population of GPs and specialists in 2018.**

| | Population of GPs and specialists (2018) | | | Trial participants | | |
|---|---|---|---|---|---|---|
| | Mean | SD | n | Mean | SD | n[a] |
| Specialist (%) | 48.8 | 50.0 | 43892 | 48.0 | 50.0 | 18237 |
| Male (%) | 64.0 | 48.0 | 43887 | 56.9 | 49.5 | 18239 |
| Age | 51.8 | 11.9 | 43392 | 50.0 | 12.4 | 18037 |
| <35 yrs old | 7.6 | 26.6 | 43392 | 12.9 | 33.5 | 18037 |
| 35–39 yrs old | 9.0 | 28.6 | 43392 | 9.9 | 29.9 | 18037 |
| 40–44 yrs old | 14.2 | 34.9 | 43392 | 14.5 | 35.2 | 18037 |
| 45–49 yrs old | 14.5 | 35.2 | 43392 | 13.6 | 34.3 | 18037 |
| 50–54 yrs old | 14.1 | 34.8 | 43392 | 12.6 | 33.2 | 18037 |
| 55–59 yrs old | 13.4 | 34.1 | 43392 | 12.5 | 33.1 | 18037 |
| 60–64 yrs old | 11.3 | 31.7 | 43392 | 10.4 | 30.5 | 18037 |
| 65–69 yrs old | 7.6 | 26.5 | 43392 | 6.9 | 25.4 | 18037 |
| 70+ yrs old | 8.1 | 27.3 | 43392 | 6.7 | 25.1 | 18037 |
| Australian Capital Territory | 1.7 | 13.0 | 43892 | 1.7 | 12.9 | 18244 |
| New South Wales | 32.7 | 46.9 | 43892 | 28.0 | 44.9 | 18244 |
| Northern Territory | 0.7 | 8.1 | 43892 | 1.1 | 10.5 | 18244 |
| Queensland | 20.1 | 40.1 | 43892 | 20.2 | 40.1 | 18244 |
| South Australia | 7.4 | 26.2 | 43892 | 7.5 | 26.3 | 18244 |
| Tasmania | 2.2 | 14.6 | 43892 | 2.7 | 16.2 | 18244 |
| Victoria | 25.2 | 43.4 | 43892 | 28.1 | 44.9 | 18244 |
| Western Australia | 10.0 | 30.0 | 43892 | 10.8 | 31.0 | 18244 |
| MMM1 (Major cities) | 80.5 | 39.6 | 42967 | 74.9 | 43.4 | 18244 |
| MMM2 | 8.3 | 27.5 | 42967 | 9.9 | 29.9 | 18244 |
| MMM3 | 5.9 | 23.5 | 42967 | 7.2 | 25.8 | 18244 |
| MMM4 | 2.3 | 14.9 | 42967 | 2.8 | 16.5 | 18244 |
| MMM5 | 2.5 | 15.7 | 42967 | 3.9 | 19.3 | 18244 |
| MMM6 | 0.4 | 6.6 | 42967 | 1.2 | 10.7 | 18244 |
| MMM7 | 0.1 | 3.5 | 42967 | 0.2 | 4.7 | 18244 |
| SEIFA Q1 (High SES) | 21.0 | 40.7 | 43679 | 21.6 | 41.2 | 18131 |
| SEIFA Q2 | 19.6 | 39.7 | 43679 | 19.9 | 39.9 | 18131 |
| SEIFA Q3 | 20.0 | 40.0 | 43679 | 20.0 | 40.0 | 18131 |
| SEIFA Q4 | 20.1 | 40.1 | 43679 | 19.7 | 39.8 | 18131 |
| SEFIA Q5 (Low SES) | 19.2 | 39.4 | 43679 | 18.8 | 39.1 | 18131 |
| Percent of popn under 5 yrs | 5.694 | 1.617 | 43728 | 5.665 | 1.599 | 18151 |
| Percent of popn above 65 yrs | 13.160 | 4.571 | 43728 | 13.403 | 4.697 | 18151 |

a. This is slightly lower than 18,237 of trial participants due to some missing values of characteristics.

physician populations approached initially by mail, or where it is unknown a priori if they have an email address. The ATET of a 2.37 percentage point fall in the response rate is relevant to physician populations where only an email address is available to researchers, and so for physicians willing to supply an email address. The fall in response rate is higher for GPs than for non-GP specialists. The effect size seems quite small possibly because the control group, though approached by mail, still had the option of completing the survey online given the mixed mode of survey completion available to them. Our estimates are therefore conservative compared to using mail by itself. Our results are, more relevant to surveys using mixed modes of delivery.

**Table 2. Characteristics of participants in intervention and control groups.**

| | Intervention | Control | N (Intervention) | N (Control) |
|---|---|---|---|---|
| Specialist (n, %) | 4376 (48.0) | 4377 (48.0) | 9122 | 9125 |
| Male (n, %) | 5164 (56.6) | 5220 (57.2) | 9118 | 9120 |
| Age (mean, sd) | 49.9 (12.5) | 49.8 (12.5) | 8423 | 8349 |
| *Age categories* (n, %) | | | 8423 | 8349 |
| <35 yrs old | 1129 (13.4) | 1158 (13.9) | | |
| 35–39 yrs old | 849 (10.1) | 846 (10.1) | | |
| 40–44 yrs old | 1239 (14.7) | 1195 (14.3) | | |
| 45–49 yrs old | 1118 (13.3) | 1110 (13.3) | | |
| 50–54 yrs old | 1052 (12.5) | 999 (12.0) | | |
| 55–59 yrs old | 1015 (12.1) | 1063 (12.7) | | |
| 60–64 yrs old | 869 (10.3) | 862 (10.3) | | |
| 65–69 yrs old | 570 (6.8) | 577 (6.9) | | |
| 70+ yrs old | 582 (6.9) | 539 (6.5) | | |
| *State/Territory* (n, %) | | | 9120 | 9124 |
| Australian Capital Territory | 147 (1.6) | 163 (1.8) | | |
| New South Wales | 2523 (27.7) | 2588 (28.4) | | |
| Northern Territory | 100 (1.1) | 103 (1.1) | | |
| Queensland | 1816 (19.9) | 1865 (20.4) | | |
| South Australia | 690 (7.6) | 673 (7.4) | | |
| Tasmania | 248 (2.7) | 246 (2.7) | | |
| Victoria | 2557 (28.0) | 2561 (28.1) | | |
| Western Australia | 1039 (11.4) | 925 (10.1) | | |
| *Rurality* (n, %) | | | 9120 | 9124 |
| MMM1 (Major cities) | 6815 (74.7) | 6844 (75.0) | | |
| MMM2 | 905 (9.9) | 903 (9.9) | | |
| MMM3 | 652 (7.1) | 658 (7.2) | | |
| MMM4 | 261 (2.9) | 250 (2.7) | | |
| MMM5 | 354 (3.9) | 350 (3.8) | | |
| MMM6 | 113 (1.2) | 98 (1.1) | | |
| MMM7 | 20 (0.2) | 21 (0.2) | | |
| *Socio-economic status of postcode* (n, %) | | | 9059 | 9072 |
| SEIFA Q1 (High SES) | 1855 (20.5) | 1873 (20.6) | | |
| SEIFA Q2 | 1807 (19.9) | 1729 (19.1) | | |
| SEIFA Q3 | 1766 (19.5) | 1859 (20.5) | | |
| SEIFA Q4 | 1882 (20.8) | 1784 (19.7) | | |
| SEFIA Q5 (Low SES) | 1749 (19.3) | 1827 (20.1) | | |
| Percent of popn. under 5 yrs (mean, sd) | 5.7 (1.6) | 5.7 (1.6) | 9067 | 9084 |
| Percent of popn. above 65 yrs (mean, sd) | 13.4 (4.7) | 13.4 (4.7) | 9067 | 9084 |
| Boost sample (n, %) | 1501 (16.5) | 1502 (16.5) | 9122 | 9125 |
| Continuing doctor (n, %) | 6690 (73.3) | 6692 (73.3) | 9122 | 9125 |
| Received incentive cheque (n, %) | 127 (1.4) | 114 (1.2) | 9122 | 9125 |
| Online completion (n, %) | 1638 (52.0) | 1515 (46.1) | 3152 | 3286 |

Weaver et al. (2019) randomised around 1,200 physicians in Minnesota and found a web-only response rate of 15.2% compared to 18.9% in the mail-only mode (3.7 percentage points), slightly higher than in our study. They also compared mixed modes (web-mail and mail-web) so the sample sizes in the web-only and mail-only groups were relatively small and so this

**Table 3. Characteristics of control and intervention group: Participants who supplied and an email address.**

| | Intervention group: have email and sent email | Control group: have email but were sent mail | N (Intervention) | N (Control) |
|---|---|---|---|---|
| Specialist (n, %) | 1904 (50.6) | 1,857 (50.5) | 3765 | 3674 |
| Male (n, %) | 2092 (55.6) | 2,807 (56.8) | 3765 | 3673 |
| Age (mean, sd) | 48.8 (11.3) | 48.7 (11.6) | 3728 | 3613 |
| *Age categories* (n, %) | | | 3728 | 3613 |
| <35 yrs old | 457 (12.3) | 489 (13.53) | | |
| 35–39 yrs old | 427 (11.5) | 423 (11.7) | | |
| 40–44 yrs old | 624 (16.7) | 578 (16.0) | | |
| 45–49 yrs old | 561 (15.0) | 537 (14.9) | | |
| 50–54 yrs old | 483 (13.0) | 431 (11.9) | | |
| 55–59 yrs old | 440 (11.8) | 449 (12.4) | | |
| 60–64 yrs old | 359 (9.6) | 335 (9.3) | | |
| 65–69 yrs old | 218 (5.8) | 210 (5.8) | | |
| 70+ yrs old | 159 (4.3) | 161 (4.5) | | |
| *State/Territory* (n, %) | | | 3765 | 3674 |
| Australian Capital Territory | 65 (1.7) | 71 (1.9) | | |
| New South Wales | 992 (26.3) | 1021 (27.8) | | |
| Northern Territory | 40 (1.1) | 46 (1.3) | | |
| Queensland | 750 (19.9) | 759 (20.1) | | |
| South Australia | 240 (6.4) | 257 (7.0) | | |
| Tasmania | 85 (2.3) | 94 (2.6) | | |
| Victoria | 1114 (29.6) | 1015 (27.6) | | |
| Western Australia | 479 (12.7) | 411 (11.2) | | |
| *Rurality* (n, %) | | | 3765 | 3674 |
| MMM1 (Major cities) | 2827 (75.1) | 2769 (75.4) | | |
| MMM2 | 364 (9.7) | 364 (9.9) | | |
| MMM3 | 269 (7.1) | 264 (7.2) | | |
| MMM4 | 106 (2.8) | 106 (2.9) | | |
| MMM5 | 143 (3.8) | 126 (3.4) | | |
| MMM6 | 45 (1.2) | 33 (0.9) | | |
| MMM7 | 11 (0.3) | 12 (0.3) | | |
| *Socio-economic status of postcode* (n, %) | | | 3732 | 3655 |
| SEIFA quintile 1 | 778 (20.9) | 718 (19.6) | | |
| SEIFA quintile 2 | 724 (19.4) | 701 (19.2) | | |
| SEIFA quintile 3 | 718 (19.2) | 786 (21.5) | | |
| SEIFA quintile 4 | 769 (20.6) | 696 (19.0) | | |
| SEFIA quintile 5 | 743 (19.9) | 754 (20.6) | | |
| Percent of popn. under 5 yrs (mean, sd) | 5.6 (1.6) | 5.6 (1.6) | 3737 | 3658 |
| Percent of popn. above 65 yrs (mean, sd) | 13.3 (4.7) | 13.4 (4.7) | 3737 | 3658 |
| Boost sample (n, %) | 74 (2.0) | 100 (2.7) | 3765 | 3674 |
| Continuing doctor (n, %) | 3605 (95.8) | 3500 (95.3) | 3765 | 3500 |
| Received incentive cheque (n, %) | 49 (1.3) | 40 (1.1) | 3765 | 3674 |
| Online completion (n, %) | 533 (71.3) | 399 (48.1) | 748 | 830 |

difference was not statistically significant. Beebe et al., (2018) randomised 686 physicians, nurses and physician assistants and found a response rate of 38.2% for web only compared to 32.1% for mail only after two reminders, a slightly higher effect size (4.1 percentage points) and in the opposite direction of our results. However, again this difference was not statistically significant.

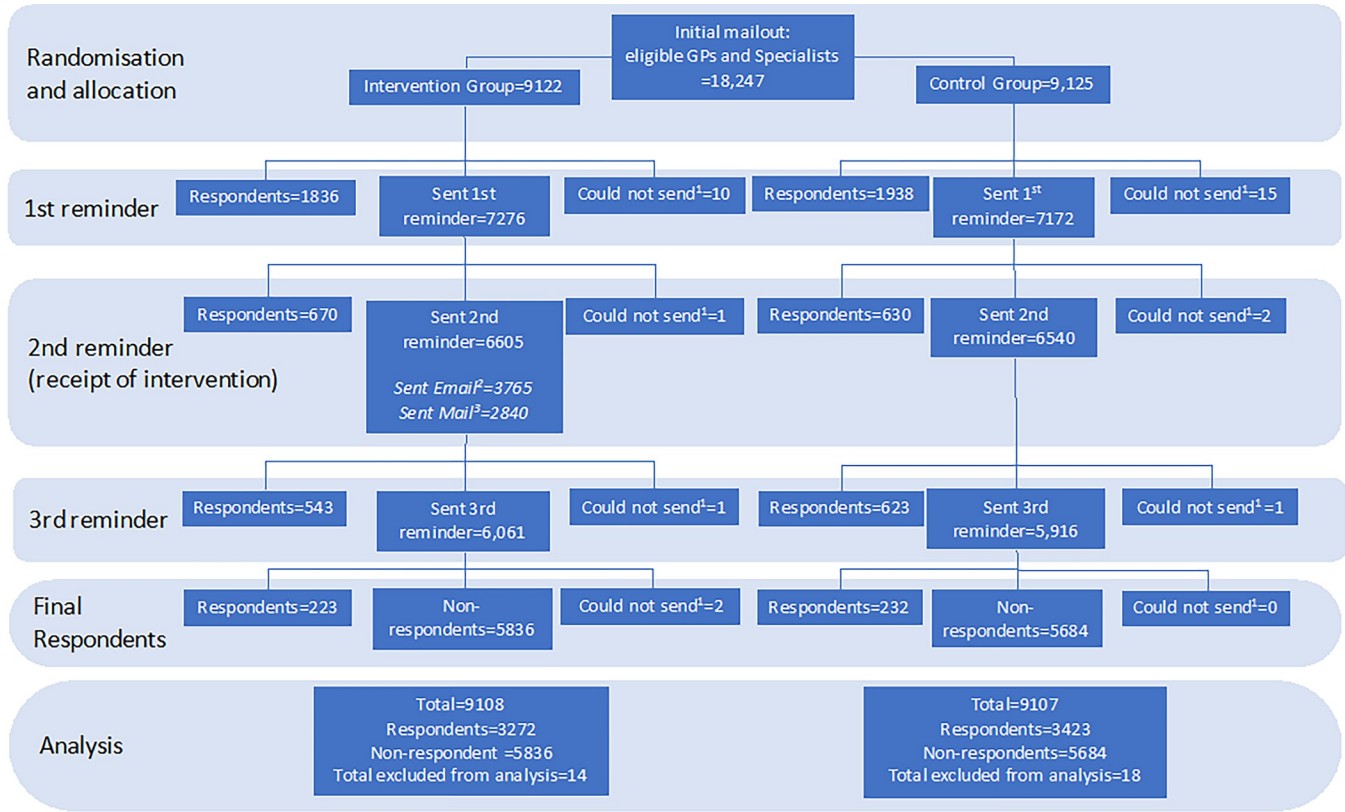

Notes. 1. Deceased or could not be sent by AMPCo because of invalid address. 2. Number with a valid email address. 3. No email on record and so sent mail. Includes one respondent with an email address but who wanted to be sent mail.

**Fig 1. Flow chart.**

The results should also be interpreted in the context of our longitudinal study that the first 10 annual waves combined a mixed mode of completion with a single method of approach using a mailed paper letter for the main mailout and three reminders. The intervention changed the method of approach to email for the second reminder. We used the second reminder, rather than the first mailout, to avoid a potentially large fall in the total number of responses if email was worse than mail. In the second reminder fewer respondents in total would be included in the trial. We do not think that the percentage difference in response rates would be any different if the intervention was for the first mailing or the first or the third reminder. At the second reminder, the intervention group were approached by email and could only complete the survey online, whilst the control group were approached by mailed letter that included a paper copy of the survey and so had a choice of online or paper copy completion. A limitation is that we are not just comparing differences in the method of approach. Though the control group were approached by mail, they could choose online or paper completion, and so their response rate might be lower if they could complete only online or higher if they could only complete using paper. For those who responded, there is evidence that those in the intervention group were more likely to complete the survey online, presumably because this group were approached by email in the second reminder. In the ITT analysis, 46.1% of control group responded online compared to 52.0% in the intervention group (Table 1). In the ATET analysis of those who supplied an email address, 48.1% in the control group completed the survey online compared to 71.3% in the intervention group.

**Table 4. Comparison of response rates.**

| | All doctors | GPs | Non-GP specialists |
|---|---|---|---|
| **Intention to treat analysis[a]** | | | |
| **Unadjusted analysis** | | | |
| Control (# responded/# invited, %) | 3423/9107 (37.59) | 1616/4734 (34.14) | 1807/4373 (41.32) |
| Intervention (# responded/# invited, %) | 3272/9108 (35.92) | 1487/4739 (31.38) | 1785/4369 (40.86) |
| Pearson $\chi^2(1)$ | 5.4 (p = 0.020) | 8.2 (p = 0.004) | 0.20 (p = 0.658) |
| Odds ratio | 0.931** | 0.882*** | 0.981 |
| (95% CI) | (0.877 to 0.989) | (0.810 to 0.961) | (0.901 to 1.07) |
| Difference in response rate (percentage points, 95% CI) | -1.66** | -2.76*** | -0.47 |
| (Intervention minus control) | (-3.06 to -0.262) | (-4.65 to -0.87) | (-2.53 to 1.60) |
| **Adjusted analysis[b]** | | | |
| Odds ratio | 0.916*** | 0.879*** | 0.936 |
| (95% CI) | (0.859 to 0.977) | (0.802 to 0.964) | (0.853 to 1.03) |
| Difference in response rate (percentage points, 95% CI) | -1.93*** | -2.52*** | -1.56 |
| (Intervention minus control) | (-3.36 to -0.50) | (-4.33 to -0.72) | (-3.80 to 0.67) |
| **Average treatment effect on the treated (ATET)[a]** | | | |
| **Unadjusted analysis** | | | |
| Control (# responded/# invited, %) | 847/3672 (23.07) | 417/1816 (22.96) | 430/1856 (23.17) |
| Intervention (# responded/# invited, %) | 769/3763 (20.44) | 368/1860 (19.78) | 401/1903 (21.07) |
| Pearson $\chi^2(1)$ | 7.7 (p = 0.006) | 5.5 (p = 0.019) | 2.4 (p = 0.122) |
| Odds ratio | 0.857*** | 0.827** | 0.885 |
| (95% CI) | (0.767 to 0.957) | (0.707 to 0.969) | (0.759 to 1.03) |
| Difference in response rate (percentage points, 95% CI) | -2.63*** | -3.17** | -2.10*** |
| (Intervention minus control) | (-4.50 to -0.75) | (-5.83 to -0.53) | (-4.75 to 0.56) |
| **Adjusted analysis[1]** | | | |
| Odds ratio | 0.847*** | 0.832** | 0.859* |
| (95% CI) | (0.750 to 0.957) | (0.695 to 0.995) | (0.726 to 1.01) |
| Difference in response rate (percentage points, 95% CI) | -2.37*** | -2.50** | -2.33* |
| (Intervention minus control) | (-4.11 to -0.63) | (-4.93 to -0.07) | (-4.91 to 0.25) |

a. ITT: Intention to Treat. ATET: Average Treatment Effect on the Treated.

b. Adjusted analyses are based on logistic regression including all independent variables in Table 1 (except for mode of completion) and have slightly smaller sampler sample sizes because of missing values for some independent variables.

*0.05<p< = 0.10

**0.01<p< = 0.05

***p< = 0.01.

The study was conducted exclusively within the Australian context, which may limit the generalizability of the findings to other countries with different healthcare systems, professional cultures, and survey response patterns. It would be beneficial to conduct similar studies in other countries to better understand how the intervention is performed in different settings. In addition, we have not examined differences in non-response bias which could arise if those who are more likely to respond to email are different in unobserved ways and this changes the composition of the doctors who respond, e.g. younger doctors are likely to respond to email. Our previous research in the same context has shown mixed modes (mail-online and online-mail) showed evidence of response bias, and that young, male doctors working in remote areas are more likely to complete the online survey [11,23].

Our definition of a complete response of at least one question answered from Section B is quite conservative compared to what is generally recommended in the literature [24]. However, some responses to current working status (Section A in the survey), such as 'retired' or 'not working in clinical practice' meant that respondents were not required to complete the rest of the survey or were directed to complete only certain sections of the survey, e.g demographics. Nevertheless, for respondents who answered at least one question from Section B, item response rates are over 90% for all sections of the survey across all 11 waves [18].

As we were uncertain about the impact of email on response rates, we took a cautious approach by not using email in the first and main mailout but chose to use email in the second reminder where potential adverse impacts on overall response rate would be minimised. Our results are therefore conservative estimates of the impact of using an email approach versus mailed paper letter. If the intervention was delivered during the main mailout, the statistical power would have been higher. Although we do have a pilot survey every year and could have tested our hypothesis using the pilot sample of around 2000 doctors, the sample size in the pilot was not large enough and most pilot responses are merged into in the main wave if the surveys do not change so still count towards the main response rate.

## Conclusions

The use of an emailed approach and online completion avoids printing and postage costs and the costs of manual data entry for those who responded to the second reminder. However, the lower response rates means that costs increase for the third mailed reminder as a higher number of participants needed to be approached. It remains unclear what advice to provide researchers as one might be willing to accept a lower response rate if costs are also lower. Often budgets for such surveys are very limited and email is the only option. In this case, optimising other survey characteristics (e.g. survey length) that can increase response rates is important using existing evidence [1,25] and new research in the context of physicians. In addition, qualitative research is helpful to better understand response behaviours [23]. However, we do recommend that researchers attempt to negotiate larger budgets for physician surveys to ensure that mailed paper letters are used where possible so that survey responses are externally valid. This is necessary to be able to make valid recommendations from survey research. Further research should test a mix of methods to approach potential respondents, with appropriate sample size calculations, and where the same subjects are approached by email as well as mailed letter or other types of contact such as text messaging and social media, though these methods might be less effective in older physician populations.

## Acknowledgments

We thank the doctors who participated in the MABEL survey.

## Author Contributions

**Conceptualization:** Tamara Taylor, Grant Russell, Anthony Scott.

**Data curation:** Tamara Taylor.

**Formal analysis:** Benjamin Harrap, Anthony Scott.

**Funding acquisition:** Anthony Scott.

**Methodology:** Tamara Taylor, Anthony Scott.

**Project administration:** Tamara Taylor, Anthony Scott.

**Supervision:** Grant Russell, Anthony Scott.

**Writing – original draft:** Anthony Scott.

**Writing – review & editing:** Benjamin Harrap, Tamara Taylor, Grant Russell.

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
