## [Decision Letter · Decision Letter 0]

10 Apr 2023

PONE-D-22-23734A randomised controlled trial of email versus mailed invitation letter in a national longitudinal survey of physicians.PLOS ONE

Dear Dr. Scott,

Thank you for submitting your manuscript to PLOS ONE. After careful consideration, we feel that it has merit but does not fully meet PLOS ONE’s publication criteria as it currently stands. Therefore, we invite you to submit a revised version of the manuscript that addresses the points raised during the review process.

ACADEMIC EDITOR:

Thank you for submitting your paper to our journal. We appreciate the effort and time you have invested in your research. After a thorough review, we believe your study has merit and potential to contribute to the field.

In order to help you improve the quality of your manuscript, we have provided detailed feedback and suggestions. We kindly ask you to consider these comments and critiques carefully as you revise your paper. Addressing these points will not only strengthen the methodology and discussion but also enhance the overall clarity, organization, and significance of your study.

Once you have made the necessary revisions, please resubmit your manuscript for further review. We look forward to receiving your revised paper and evaluating its potential for publication in our journal.

Thank you for considering our feedback, and we hope to receive your revised manuscript soon.

We look forward to receiving your revised manuscript.

Kind regards,

Fares Alahdab, MD, MSc

Academic Editor

PLOS ONE

Journal Requirements:

Reviewers' comments:

Reviewer's Responses to Questions

**Comments to the Author**

1. Is the manuscript technically sound, and do the data support the conclusions?

Reviewer #1: No

Reviewer #2: Yes

Reviewer #3: Yes

2. Has the statistical analysis been performed appropriately and rigorously? 

Reviewer #1: No

Reviewer #2: Yes

Reviewer #3: Yes

3. Have the authors made all data underlying the findings in their manuscript fully available?

Reviewer #1: No

Reviewer #2: No

Reviewer #3: No

4. Is the manuscript presented in an intelligible fashion and written in standard English?

Reviewer #1: Yes

Reviewer #2: Yes

Reviewer #3: Yes

5. Review Comments to the Author

Reviewer #1: I have several concerns about the current study such that I cannot recommend it for publication at PLOS one.

First, the topic is about a survey method, and it is comparing existing methods (online vs. letter recruitment for survey) rather than an innovative method. The sample size is restricted to be physicians, and the effect that was found is extremely small (i.e., a difference in about 2% response rate) albeit significant, and it is confirming previous findings. Given these, I think this study is more suitable for a specialized journal on survey methods.

Second, the study was “administered between August 2018 and April 2019” I don’t know what the policy of PLOS-one is about this issue, but some journals do not accept results from surveys that are more than 3 years old. Furthermore, given that the motivation for the current study is that similar studies use more than 10-year-old data and the authors argued for an increase in internet usage. As we all know a lot of things happened since April of 2019 with emails and zoom meetings serving as essential tools in our society, so I am not sure whether the current results can be treated as up-to-date findings.

Third, the methods are idiosyncratic in that the main manipulation took place only during the second reminder after the recruitment took place over mails and the letter reminder. To quote the authors, “More specifically, we introduce an email approach in the second of three reminders sent to non-responding physicians. In the first ten annual waves of the survey, the main mailout and all three reminders were delivered by mail only.” What was the reason that this specific mixed method was used for the study? Are there any empirical or theoretical basis to suggest that this method would be the best? In the discussion, the authors acknowledge these as potential limitations, and I think these limitations seriously compromise the generalizability of the findings.

Fourth, it is unclear what the outcome measures; are they response rates only after the second reminder or after the first reminder? It sounds like it is the response rate regardless of the timepoint, and this can be an issue as noted in the next point.

Fifth, the results reported in the abstract and as the main findings included data from two third of the participants in the intervention group who did not even receive the experimental treatment because they did not even have valid email addresses! These data should not even be reported. The authors also report results from those who had email addresses in both the intervention and control groups, and the response rates of both groups among these people are reduced from mid- to high thirty percent of the overall response rates down to low 20’s. Could it be the case that those who provided email addresses were more likely to have responded even before the manipulation?

Other less major questions:

Was the survey anonymous? If so, how was the online completion tracked in terms of which condition they belonged? If the survey responses had identifiers, couldn’t that be a factor that might compromise the generalizability of the findings?

First sentence of Background; “Web surveys have lower response rates than other survey modes.” This is a strong statement and perhaps incorrect given the findings of the current study. Also, what are “other survey modes”?

“…and this study also included nurses and physician assistants as well as physicians” This can sound somewhat politically incorrect as if nurses and physician assistants’ results are meaningless.

Reviewer #2: The article provides an important contribution regarding surveys and data collection. It makes a good point, indicating that most of the studies in this area are ten years old and thus requires an update. It utilizes sound statistical methods and contributes to existing literature.

Reviewer #3: Overall, the paper is well-structured, and the research question is clearly defined. The authors aim to compare the response rates between email and mailed approaches within a national longitudinal survey of physicians. I agree with the authors that this paper is up to date, considering that most studies in this area use data more than 10 years old, which justifies the need to re-examine this issue. The methods used in the study are robust and well-described, following the CONSORT guidelines and using appropriate statistical methods.

The paper provides a clear comparison of response rates between the intervention and control groups, as well as subgroups of GPs and non-GP specialists. The results indicate that the response rate was lower for the email approach compared to the mailed approach. This difference was larger for GPs compared to non-GP specialists.

In conclusion, the paper contributes valuable insights to the survey methods literature for physicians, particularly in the context of the increasing use of email and the internet in medical practice. The findings have practical implications for survey design and recruitment methods targeting physicians.

Strengths:

1. The study followed the Consolidated Standards of Reporting Trials (CONSORT) guidelines, which ensures transparent and accurate reporting of the research process.

2. The research was conducted within the context of the Medicine in Australia: Balancing Employment and Life (MABEL) survey, a well-established longitudinal panel survey of all medical practitioners in Australia, providing a strong foundation for the study.

3. The sample frame for MABEL, the Medical Directory of Australia, is a national database of doctors, ensuring a comprehensive representation of the medical practitioner population.

4. The study used a randomized controlled trial with a parallel-arm design and 1:1 allocation, which allows for a rigorous comparison between the intervention and control groups.

5. The researchers employed stratified randomization to ensure that the proportions of different doctor types (GP, specialist), continuing or new, and boost sample in the intervention and control groups were the same.

6. The analysis was conducted by a researcher who was blinded to group allocation until after the analysis was complete and checked, reducing the risk of bias.

Weaknesses:

1. The study relied on data from Wave 11, which was administered between August 2018 and April 2019, limiting the generalizability of the findings to more recent years.

2. The study excluded junior doctors, who might have different response rates and preferences compared to more experienced doctors, reducing the generalizability of the findings.

3. A significant proportion of the intervention group (43%) did not have an email address and were instead approached by mailed paper letter, which might underestimate the effect of the intervention for those who received an email.

4. The intervention only differed at the second reminder, which may not have been enough to fully capture the difference in response rates between email and mailed approaches.

5. The response rate differences between the intervention and control groups might be influenced by other factors not considered in the study, such as the timing of reminders and the content of the survey.

Suggestions for improvement:

1. The authors could consider updating the study with more recent data to ensure the relevance of the findings to current survey methods and physician response rates.

2. To further assess the generalizability of the findings, the authors could include junior doctors in the sample (for future work) and analyze their response rates separately, providing insights into different age groups and experience levels.

3. The researchers could explore additional ways to improve email deliverability and reduce the proportion of participants without an email address to better assess the intervention's effectiveness.

4. The intervention could be extended to more than one reminder stage to better understand the cumulative effect of using email approaches over multiple reminders.

5. The authors could conduct sensitivity analyses to determine the impact of other factors, such as timing of reminders and survey content, on response rates in both intervention and control groups. This would provide a more comprehensive understanding of the factors influencing physician response rates.

6. Limited generalizability across different populations: The study was conducted exclusively within the Australian context, which may limit the generalizability of the findings to other countries with different healthcare systems, professional cultures, and survey response patterns. It would be beneficial to conduct similar studies in other countries to better understand how the intervention is performed in different settings.

7. Lack of examination of non-response bias: The study focused on response rates but did not explore potential non-response bias, which may arise if the doctors who chose to participate in the survey differ systematically from those who did not. It is important to investigate whether the intervention affected the composition of respondents in any way, as this may have implications for the interpretation of the survey results and their representativeness of the target population.

8. Limited exploration of factors influencing email effectiveness: The study did not investigate factors that could influence the effectiveness of email approaches, such as the timing of the email, subject lines, or the formatting of the email content. A more detailed investigation of these factors could help identify best practices for optimizing email reminders in future survey administration.

9. Single intervention approach: The study tested only one email-based intervention, which may not represent the full range of potential interventions that could be implemented to improve response rates. Future research could explore other interventions, such as varying the incentives for participation or using different communication channels, to determine the most effective strategies for increasing response rates.

10. No qualitative insights: The study did not provide any qualitative insights into the reasons behind the observed response rate differences between the intervention and control groups. Conducting interviews or focus groups with a subset of physicians could help researchers better understand the reasons for their survey participation preferences, which could inform future survey design and administration strategies.

11. Provide more information on the implications of their findings for healthcare and survey research communities.

12. Address the study's limitations in more detail and discuss potential ways to overcome them in future research.

13. Elaborate on the practical implications of their findings, such as cost savings, logistical benefits, and challenges or barriers to implementing email reminders.

14. Clearly outline the next steps needed for future research to build upon the current study and address its limitations.

Writing Quality, Understanding Easiness, Clarity, and Organization:

- Overall, the writing quality of the paper seems to be good, with clear descriptions of the methods, results, and conclusions. The organization appears logical, with sections following a conventional structure for a research article.

- Suggestions for improvement:

a) Some portions of the text, particularly in the Methods section, could be further clarified for better readability. Consider rephrasing complex sentences and providing more straightforward explanations of the study design and data analysis.

b) In the Introduction and Discussion sections, the authors could provide more context on the importance of their research question and the implications of their findings for the broader healthcare and survey research communities.

Content Accuracy, Comprehensiveness, Clinical Usefulness, and Significance:

- The content appears accurate, and the methodology is comprehensive. However, some concerns have been raised regarding the study's limitations, which could impact the clinical usefulness and significance of the findings.

- Suggestions for improvement:

a) Address the weaknesses mentioned in previous comments, such as the limited generalizability of the study, potential non-response bias, and factors influencing email effectiveness.

b) Explore additional intervention approaches, such as varying incentives for participation, using different communication channels, or incorporating personalized messaging to identify the most effective strategies for increasing response rates.

c) Consider conducting a qualitative investigation to understand the reasons behind physicians' survey participation preferences, which could inform future survey design and administration strategies.

d) Provide a more in-depth literature review to better contextualize the study within the broader research landscape and identify gaps in current knowledge.

e) Discuss the practical implications of the study findings, including potential cost savings or logistical benefits of using email reminders, as well as any challenges or barriers to implementation.

f) Include a more detailed description of the target population and the representativeness of the sample to help readers understand the context and relevance of the study findings.

g) Present a clear plan for future research, outlining the next steps needed to build upon the current study and address its limitations.

6. PLOS authors have the option to publish the peer review history of their article (what does this mean?). If published, this will include your full peer review and any attached files.

Reviewer #1: No

Reviewer #2: No

Reviewer #3: No

---

## [Author Response · Author response to Decision Letter 0]

30 Jun 2023

Response to editors comments (shown by *)

*We have made the above changes to style.

2. Please provide additional details regarding participant consent. In the ethics statement in the Methods and online submission information, please ensure that you have specified what type you obtained (for instance, written or verbal, and if verbal, how it was documented and witnessed). If your study included minors, state whether you obtained consent from parents or guardians. If the need for consent was waived by the ethics committee, please include this information.?

*We have made changes to the methods section and the inline submission form

Response to reviewers comments

Reviewer #1: I have several concerns about the current study such that I cannot recommend it for publication at PLOS one.

First, the topic is about a survey method, and it is comparing existing methods (online vs. letter recruitment for survey) rather than an innovative method. The sample size is restricted to be physicians, and the effect that was found is extremely small (i.e., a difference in about 2% response rate) albeit significant, and it is confirming previous findings. Given these, I think this study is more suitable for a specialized journal on survey methods.

*Survey methods are essential to all applied researchers, and response rates are feature of many any studies and surveys undertaking primary data collection, especially collecting data from health care providers. The size of a treatment effect should not influence publication otherwise this leads to publication bias. We think the context of our study is innovative given the increasing complexity of survey design and administration.

Second, the study was “administered between August 2018 and April 2019” I don’t know what the policy of PLOS-one is about this issue, but some journals do not accept results from surveys that are more than 3 years old. Furthermore, given that the motivation for the current study is that similar studies use more than 10-year-old data and the authors argued for an increase in internet usage. As we all know a lot of things happened since April of 2019 with emails and zoom meetings serving as essential tools in our society, so I am not sure whether the current results can be treated as up-to-date findings.

*The pandemic caused a delay in publication of this research and so we should not be disadvantaged by this. We agree that the pandemic may have changed people’s preferences towards online communication away from face to face (we now mention this at the beginning of the discussion), but both alternatives in our study (mail and email) should not have been affected by this as both do not require face to face contact. We did not compare email with the use of face-to face interviews, which might then have affected the generalisability of our results post-pandemic.

Third, the methods are idiosyncratic in that the main manipulation took place only during the second reminder after the recruitment took place over mails and the letter reminder. To quote the authors, “More specifically, we introduce an email approach in the second of three reminders sent to non-responding physicians. In the first ten annual waves of the survey, the main mailout and all three reminders were delivered by mail only.” What was the reason that this specific mixed method was used for the study? Are there any empirical or theoretical basis to suggest that this method would be the best? In the discussion, the authors acknowledge these as potential limitations, and I think these limitations seriously compromise the generalizability of the findings.

*As noted, we acknowledge these limitations in the discussion and add to the text to highlight that the percentage difference in response rates should not be any different if we implemented the intervention in the first mailout, 1st reminder or third reminder. We have also noted in the text that we used the second reminder to avoid a large fall in the total number of responses if email was worse than mail. In the second reminder fewer respondents in total would be included in the trial.

Fourth, it is unclear what the outcome measures; are they response rates only after the second reminder or after the first reminder? It sounds like it is the response rate regardless of the timepoint, and this can be an issue as noted in the next point.

*Yes the outcomes are the final response rates at the end of recruitment. We have added text to p6 to make this clear.

Fifth, the results reported in the abstract and as the main findings included data from two third of the participants in the intervention group who did not even receive the experimental treatment because they did not even have valid email addresses! These data should not even be reported. The authors also report results from those who had email addresses in both the intervention and control groups, and the response rates of both groups among these people are reduced from mid- to high thirty percent of the overall response rates down to low 20’s. Could it be the case that those who provided email addresses were more likely to have responded even before the manipulation?

*Excluding this group would bias the results and make them less generalisable to a real world setting when email addresses are not always available. This was an intention to treat analysis with the intervention occurring at the second reminder. Not having access to email is what happens in real world settings and so provides more generalisable results relevant to the population of doctors. Those who provided email addresses could be more likely to respond, but randomisation ensures the pre-intervention propensity to respond and the pre-intervention proportion with an email address are the same in the intervention and control groups.

Other less major questions:

Was the survey anonymous? If so, how was the online completion tracked in terms of which condition they belonged? If the survey responses had identifiers, couldn’t that be a factor that might compromise the generalizability of the findings?

*All MABEL survey waves were not anonymous – this is necessary in longitudinal surveys that track respondents over time. Respondents were tracked as usual as part of the longitudinal survey. For the online version of the survey, respondents were required to log in with their unique username and password. Paper versions of surveys had their username printed on the front of each survey. Text has been added/moved on p5.

First sentence of Background; “Web surveys have lower response rates than other survey modes.” This is a strong statement and perhaps incorrect given the findings of the current study. Also, what are “other survey modes”?

*This is a strong statement because it comes from a large systematic review whose results are unambiguous. Other modes include face to face, mail, telephone, and email. We have clarified the text in this sentence. 

“…and this study also included nurses and physician assistants as well as physicians” This can sound somewhat politically incorrect as if nurses and physician assistants’ results are meaningless.

*We have modified the text here.

Reviewer #2: The article provides an important contribution regarding surveys and data collection. It makes a good point, indicating that most of the studies in this area are ten years old and thus requires an update. It utilizes sound statistical methods and contributes to existing literature.

Reviewer #3: Overall, the paper is well-structured, and the research question is clearly defined. The authors aim to compare the response rates between email and mailed approaches within a national longitudinal survey of physicians. I agree with the authors that this paper is up to date, considering that most studies in this area use data more than 10 years old, which justifies the need to re-examine this issue. The methods used in the study are robust and well-described, following the CONSORT guidelines and using appropriate statistical methods.

The paper provides a clear comparison of response rates between the intervention and control groups, as well as subgroups of GPs and non-GP specialists. The results indicate that the response rate was lower for the email approach compared to the mailed approach. This difference was larger for GPs compared to non-GP specialists.

In conclusion, the paper contributes valuable insights to the survey methods literature for physicians, particularly in the context of the increasing use of email and the internet in medical practice. The findings have practical implications for survey design and recruitment methods targeting physicians.

Strengths:

1. The study followed the Consolidated Standards of Reporting Trials (CONSORT) guidelines, which ensures transparent and accurate reporting of the research process.

2. The research was conducted within the context of the Medicine in Australia: Balancing Employment and Life (MABEL) survey, a well-established longitudinal panel survey of all medical practitioners in Australia, providing a strong foundation for the study.

3. The sample frame for MABEL, the Medical Directory of Australia, is a national database of doctors, ensuring a comprehensive representation of the medical practitioner population.

4. The study used a randomized controlled trial with a parallel-arm design and 1:1 allocation, which allows for a rigorous comparison between the intervention and control groups.

5. The researchers employed stratified randomization to ensure that the proportions of different doctor types (GP, specialist), continuing or new, and boost sample in the intervention and control groups were the same.

6. The analysis was conducted by a researcher who was blinded to group allocation until after the analysis was complete and checked, reducing the risk of bias.

Weaknesses:

1. The study relied on data from Wave 11, which was administered between August 2018 and April 2019, limiting the generalizability of the findings to more recent years.

*COVID delayed the publication of this paper, and the COVID years. There is no a priori reason to suggest that the results are less relevant inn2023. COVID might have changed preferences for the use internet/email away from face to face, but our intervention did not compare with face to face survey completion.

2. The study excluded junior doctors, who might have different response rates and preferences compared to more experienced doctors, reducing the generalizability of the findings.

*Yes this correct – our findings are generalisable only to qualified GPs and specialists.

3. A significant proportion of the intervention group (43%) did not have an email address and were instead approached by mailed paper letter, which might underestimate the effect of the intervention for those who received an email.

*Yes – we have accounted for this by also conducting the analysis excluding those without email addresses (the average treatment effect on the treated) where the treatment effect is slightly higher.

4. The intervention only differed at the second reminder, which may not have been enough to fully capture the difference in response rates between email and mailed approaches.

*See response to Reviewer 1. A smaller total number of respondents are exposed to the intervention at the second reminder. We believe this would not have influenced the size of the percentage change in response rates between intervention and control, but would have led to lower power to detect a difference (pp14-15). However, our sample size calculation meant that we had sufficient power. 

5. The response rate differences between the intervention and control groups might be influenced by other factors not considered in the study, such as the timing of reminders and the content of the survey.

*This is correct - however randomisation ensured these were the same in the intervention and control groups.

Suggestions for improvement:

1. The authors could consider updating the study with more recent data to ensure the relevance of the findings to current survey methods and physician response rates.

*This is not possible as the last wave of the survey was in 2018-19.

2. To further assess the generalizability of the findings, the authors could include junior doctors in the sample (for future work) and analyze their response rates separately, providing insights into different age groups and experience levels.

*Junior doctors were not included in the randomisation as we focused only on more senior doctors. We will consider this for further research.

3. The researchers could explore additional ways to improve email deliverability and reduce the proportion of participants without an email address to better assess the intervention's effectiveness.

*Noted.

4. The intervention could be extended to more than one reminder stage to better understand the cumulative effect of using email approaches over multiple reminders.

*Yes we could have used email in all reminders as an intervention. There are various types of mixed mode that can be used.

5. The authors could conduct sensitivity analyses to determine the impact of other factors, such as timing of reminders and survey content, on response rates in both intervention and control groups. This would provide a more comprehensive understanding of the factors influencing physician response rates.

*We have another paper more generally examining factors influencing response rates. Our study was focused on the specific intervention and other factors were held constant because of randomisation.

Taylor T, Scott A. Do Physicians Prefer to Complete Online or Mail Surveys? Findings From a National Longitudinal Survey. Evaluation & the Health Professions. 2018;42(1):41-70. doi: 10.1177/0163278718807744

6. Limited generalizability across different populations: The study was conducted exclusively within the Australian context, which may limit the generalizability of the findings to other countries with different healthcare systems, professional cultures, and survey response patterns. It would be beneficial to conduct similar studies in other countries to better understand how the intervention is performed in different settings.

*We have included this as a limitation in the discussion on p15.

7. Lack of examination of non-response bias: The study focused on response rates but did not explore potential non-response bias, which may arise if the doctors who chose to participate in the survey differ systematically from those who did not. It is important to investigate whether the intervention affected the composition of respondents in any way, as this may have implications for the interpretation of the survey results and their representativeness of the target population.

*This is an important issue if those who prefer email differ systematically from those who do not, and if these factors are also correlated with specific study outcomes (e.g. difference in job satisfaction for example). We have acknowledged this possibility but also cited evidence form our previous research using MABEL that examined response rates and response bias (p15). This found that response bias could be an issue depending on the specific study objectives. 

8. Limited exploration of factors influencing email effectiveness: The study did not investigate factors that could influence the effectiveness of email approaches, such as the timing of the email, subject lines, or the formatting of the email content. A more detailed investigation of these factors could help identify best practices for optimizing email reminders in future survey administration.

*This was not an objective of our study, but we have noted this in the conclusion as an option for further research (p16).

9. Single intervention approach: The study tested only one email-based intervention, which may not represent the full range of potential interventions that could be implemented to improve response rates. Future research could explore other interventions, such as varying the incentives for participation or using different communication channels, to determine the most effective strategies for increasing response rates.

*As above – we have acknowledged this in the conclusions as an area of further research (p16).

10. No qualitative insights: The study did not provide any qualitative insights into the reasons behind the observed response rate differences between the intervention and control groups. Conducting interviews or focus groups with a subset of physicians could help researchers better understand the reasons for their survey participation preferences, which could inform future survey design and administration strategies.

*We have included this as further research in the conclusion. We did explore qualitative insights in a previous study which we have now referenced. Do Physicians Prefer to Complete Online or Mail Surveys? Findings From a National Longitudinal Survey. T. Taylor and A. Scott. Evaluation & the Health Professions 2019 Vol. 42 Issue 1 Pages 41-70. DOI: 10.1177/0163278718807744

11. Provide more information on the implications of their findings for healthcare and survey research communities.

*We have added text (as above) to the conclusion.

12. Address the study's limitations in more detail and discuss potential ways to overcome them in future research.

* We have included this in the discussion in response to all of the previous comments.

13. Elaborate on the practical implications of their findings, such as cost savings, logistical benefits, and challenges or barriers to implementing email reminders.

*This suggestion is potentially useful but is not directly related to the aims of our research.

14. Clearly outline the next steps needed for future research to build upon the current study and address its limitations.

We have done this on the basis of previous suggestions by this referee – in the discussion and conclusion.

Writing Quality, Understanding Easiness, Clarity, and Organization:

- Overall, the writing quality of the paper seems to be good, with clear descriptions of the methods, results, and conclusions. The organization appears logical, with sections following a conventional structure for a research article.

- Suggestions for improvement:

a) Some portions of the text, particularly in the Methods section, could be further clarified for better readability. Consider rephrasing complex sentences and providing more straightforward explanations of the study design and data analysis.

*We have edited the methods section to clarify things where possible, though with no specific guidance from the reviewer we hope we have addressed this concern.

b) In the Introduction and Discussion sections, the authors could provide more context on the importance of their research question and the implications of their findings for the broader healthcare and survey research communities.

*We have added text in the discussion and conclusions

Content Accuracy, Comprehensiveness, Clinical Usefulness, and Significance:

- The content appears accurate, and the methodology is comprehensive. However, some concerns have been raised regarding the study's limitations, which could impact the clinical usefulness and significance of the findings.

- Suggestions for improvement:

a) Address the weaknesses mentioned in previous comments, such as the limited generalizability of the study, potential non-response bias, and factors influencing email effectiveness.

*Done – see above responses

b) Explore additional intervention approaches, such as varying incentives for participation, using different communication channels, or incorporating personalized messaging to identify the most effective strategies for increasing response rates.

*Mentioned as further research in the conclusion

c) Consider conducting a qualitative investigation to understand the reasons behind physicians' survey participation preferences, which could inform future survey design and administration strategies.

Mentioned as further research in the conclusion

d) Provide a more in-depth literature review to better contextualize the study within the broader research landscape and identify gaps in current knowledge.

*We are limited on space and have kept the literature review focussed and specific to the research questions we are addressing.

e) Discuss the practical implications of the study findings, including potential cost savings or logistical benefits of using email reminders, as well as any challenges or barriers to implementation.

*This is discussed in the conclusion.

f) Include a more detailed description of the target population and the representativeness of the sample to help readers understand the context and relevance of the study findings

*We have included a new Table 1 comparing the respondents in trial to the population of GPs and specialists in 2018. We have included a few sentences in the text on p8 to describe this table.

g) Present a clear plan for future research, outlining the next steps needed to build upon the current study and address its limitations.

*Mentioned as further research in the conclusion.

---

## [Decision Letter · Decision Letter 1]

24 Jul 2023

A randomised controlled trial of email versus mailed invitation letter in a national longitudinal survey of physicians.

PONE-D-22-23734R1

Dear Dr. Scott,

We’re pleased to inform you that your manuscript has been judged scientifically suitable for publication and will be formally accepted for publication once it meets all outstanding technical requirements.

Kind regards,

Fares Alahdab, MD, MSc

Academic Editor

PLOS ONE

Additional Editor Comments (optional):

Reviewers' comments:

Reviewer's Responses to Questions

**Comments to the Author**

1. If the authors have adequately addressed your comments raised in a previous round of review and you feel that this manuscript is now acceptable for publication, you may indicate that here to bypass the “Comments to the Author” section, enter your conflict of interest statement in the “Confidential to Editor” section, and submit your "Accept" recommendation.

Reviewer #3: All comments have been addressed

2. Is the manuscript technically sound, and do the data support the conclusions?

Reviewer #3: Yes

3. Has the statistical analysis been performed appropriately and rigorously? 

Reviewer #3: Yes

4. Have the authors made all data underlying the findings in their manuscript fully available?

Reviewer #3: No

5. Is the manuscript presented in an intelligible fashion and written in standard English?

Reviewer #3: Yes

6. Review Comments to the Author

Reviewer #3: Thank you for addressing my comments and revising your manuscript accordingly. I have no further comments to add.

7. PLOS authors have the option to publish the peer review history of their article (what does this mean?). If published, this will include your full peer review and any attached files.

Reviewer #3: No

---

## [Editor Report · Acceptance letter]

14 Aug 2023

PONE-D-22-23734R1 

A randomised controlled trial of email versus mailed invitation letter in a national longitudinal survey of physicians. 

Dear Dr. Scott:

I'm pleased to inform you that your manuscript has been deemed suitable for publication in PLOS ONE. Congratulations! Your manuscript is now with our production department. 

Kind regards, 

on behalf of

Dr. Fares Alahdab 

Academic Editor

PLOS ONE